# Patterns of non-communicable comorbidities at start of tuberculosis treatment in three regions of the Philippines: The St-ATT cohort

**Sharon E. Cox**[1,2,3]*, **Tansy Edwards**[2,4], **Benjamin N. Faguer**[2☯], **Julius P. Ferrer**[5☯], **Shuichi J. Suzuki**[2☯], **Mitsuki Koh**[2], **Farzana Ferdous**[2], **Naomi R. Saludar**[6], **Anna-Marie C. G. Garfin**[7], **Mary C. Castro**[5], **Juan A. Solon**[5]

**1** Faculty of Epidemiology & Population Health, London School of Hygiene & Tropical Medicine, London, United Kingdom, **2** School of Tropical Medicine & Global Health, Nagasaki University, Nagasaki, Japan, **3** Institute of Tropical Medicine, Nagasaki University, Nagasaki, Japan, **4** International Statistics and Epidemiology Group, Faculty of Epidemiology & Population Health, London School of Hygiene & Tropical Medicine, London, United Kingdom, **5** Nutrition Center Philippines, Muntinlupa City, Manila, Philippines, **6** San Lazaro Hospital, Santa Cruz, Manila, Philippines, **7** Department of Health, National TB Control Programme, Manila, the Philippines

☯ These authors contributed equally to this work.
\* Sharon.Cox@LSHTM.AC.UK

**Data Availability Statement:** For the data availability, we have been advised by the LSHTM Research Data Manager that the anonymised data set should be made available on reasonable

## Abstract

Diabetes and undernutrition are common risk factors for tuberculosis (TB), associated with poor treatment outcomes and exacerbated by TB. Limited data exist describing patterns and risk factors of multiple comorbidities in persons with TB. Nine-hundred participants (69.6% male) were enrolled in the **S**tarting **A**nti-**T**B **T**reatment (St-ATT) cohort, including 133 (14.8%) initiating treatment for multi-drug resistant TB (MDR-TB). Comorbidities were defined as: diabetes, HbA1c $\geq$6.5% and/or on medication; hypertension, systolic blood pressure $\geq$140 mmHg or diastolic blood pressure $\geq$90 mmHg and/or on medication; anaemia (moderate/severe), haemoglobin <11g/dL; and, undernutrition (moderate/severe) body-mass-index <17 kg/m$^2$. The most common comorbidities were undernutrition 23.4% (210/899), diabetes 22.5% (199/881), hypertension 19.0% (164/864) and anaemia 13.5% (121/899). Fifty-eight percent had $\geq$1 comorbid condition (496/847), with 17.1% having $\geq$2; most frequently diabetes and hypertension (N = 57, 6.7%). Just over half of diabetes (54.8%) and hypertension (54.9%) was previously undiagnosed. Poor glycemic control in those on medication (HbA1c$\geq$8.0%) was common (N = 50/73, 68.5%). MDR-TB treatment was associated with increased odds of diabetes (Adjusted odds ratio (AOR) = 2.48, 95% CI: 1.55–3.95); but decreased odds of hypertension (AOR = 0.55, 95% CI: 0.39–0.78). HIV infection was only associated with anaemia (AOR = 4.51, 95% CI: 1.01–20.1). Previous TB treatment was associated with moderate/severe undernutrition (AOR = 1.98, 95% CI: 1.40–2.80), as was duration of TB-symptoms before starting treatment and household food insecurity. No associations for sex, alcohol or tobacco use were observed. MDR-TB treatment was marginally associated with having $\geq$2 comorbidities (OR = 1.52, 95% CI: 0.97–2.39). TB treatment programmes should plan for large proportions of persons requiring diagnosis and management of comorbidities with the potential to adversely affect TB treatment outcomes and quality of life. Dietary advice and nutritional management are components of comprehensive care for

request rather than completely openly accessible. It will be deposited ini LSHTM data compass with the following DOI https://doi.org/10.17037/DATA.00002476. The reason for this is that the individual informed consent stated "This anonymized data and any remaining sputum/blood samples will be available for use by other health researchers for ethically approved projects related to TB to maximize benefit from your contribution to research on the health and wellbeing of Filipinos". Hence we need to check that the data is being used as we indicated above.

**Funding:** This study was supported by funding from Nagasaki University and a grant in aid of scientific research ("Kakenhi": 17H04662), Ministry of Education, Culture, Sports, Science & Technology, Japan awarded to S Cox. The funders had no role in the design, conduct of the study or decision to publish.

**Competing interests:** All authors declare no competing interests

the above conditions as well as TB and should be included in planning of patient-centred services.

## Introduction

Tuberculosis (TB) remains the leading cause of death globally from an infectious disease with strong poverty-associated social determinants including malnutrition [1, 2]. Philippines is a high burden country for both drug-sensitive and drug-resistant TB, with one of the highest estimated incidence rates (554/100,000) globally in the context of low incidence of HIV [2]. Many low- and middle-income countries, similar to Philippines, are undergoing nutrition transition with rapidly increasing nutrition-related non-communicable diseases (NCDs) such as diabetes and hypertension, not limited to higher income groups and associated with poor-quality but energy-dense diets, sedentary lifestyles and other behavioural risk factors such as alcohol and tobacco use, also risk factors for TB [3, 4].

Undernutrition is an important risk factor for developing TB disease [5], can also result from its physiological and socio-economic consequences [6] and is associated with adverse TB treatment outcomes including death [7]. Diabetes is also a risk factor for developing active TB disease and adverse treatment outcomes including death and relapse or recurrent TB [8–10]. In the Philippines undernutrition and diabetes are estimated to be the leading population level risk factors for TB, estimated to affect 13.3% and 7.1% of adults [3, 11]. There is an increasing recognition of the burden and impact of multimorbidity (the co-existence of 2 or more medical conditions) [12] for TB and other health outcomes, but also of opportunities within TB programmes for increasing screening and uptake of NCD health services [13]. However, despite many NCDs and TB sharing nutrition-related risk factors and management strategies, consideration of nutrition screening and linkage with nutrition services for TB and NCDs is rarely considered.

Limited systematic data of multimorbidity in TB exists and often relies on retrospective data, which due to high levels of undiagnosed NCDs and lack of routine screening or data capture/reporting (e.g., body mass index (BMI)) are probable under-estimates. Furthermore, reports often focus on only one condition, most often diabetes [14] and more recently depression or common mental disorders [15]. A recent secondary analysis of WHO Global Health Survey data (collected in 2003) indicates that up to two-thirds of people with TB may have one or more NCD [16]. However, these data included NCDs based on self-report and TB disease based on self-reported TB symptoms of cough and or haemoptysis. Moreover, this survey did not collect information on hypertension or nutritional status (under- or over-weight) or anaemia. Previously, within a cross-sectional study of persons on TB treatment we reported that up to 40% of Filipino persons registered for TB treatment had at least one comorbidity of diabetes, moderate or severe anaemia (haemoglobin <11g/dl) or moderate or severe undernutrition (Body Mass Index (BMI) <17 kg/m$^2$) [17].

For implementation of the END TB strategy, it is recommended that "All persons with TB need to be assessed for nutritional status and receive nutritional counselling and care according to need" and additionally, "all persons with TB should be screened for diabetes" and, "in addition to HIV/AIDS, other co-morbidities and health risks associated with TB require integrated management" [18]. However, to plan the delivery of such services, integrated within predominantly vertical, donor-led TB programmes, better data is needed to quantify the extent of multi-morbidities experienced by persons with TB.

The aims of this cross-sectional study were to quantify the prevalence, patterns and identify predictors of non-communicable comorbidities of Filipinos newly starting TB treatment enrolled in the Starting Anti-TB Treatment (St-ATT) cohort in three regions of the Philippines.

## Methods

### Study design

A cross-sectional analysis of data collected at enrolment into a facility-based prospective cohort study conducted in public (government) TB Directly Observed Treatment (DOT) clinics in the Philippines (ISRCTN16347615).

### Setting

Participants were enrolled from 13 TB-DOTS clinics in Metro Manila (N = 3), Cebu (Region VII, N = 5) and Negros Occidental (Region VI, N = 5), including 4 centers for the programmatic management of drug resistant TB (PMDT) centers which were also HIV referral centers (2 in Negros Occidental, 1 In Metro Manila and 1 in Cebu. Metro Manila is identified as a high HIV category area compared to other areas of the country. Clinics in Cebu and Negros included urban, peri-urban and rural catchment areas (**S1 Fig**).

### Participants

All non-pregnant adults (≥18 years) registered at participating clinics and within 5 days of starting their TB treatment for pulmonary TB (bacteriologically confirmed or clinically diagnosed) on either standard treatment or on the WHO shorter regimen for multi-drug resistant TB (MDR-TB) were potentially eligible to participate. Exclusion criteria included current imprisonment, non-pulmonary TB, resident outside of study-designated barangays (village level administrative areas; some were excluded due to safety/transport limitations for field staff) or a medical or psychiatric disorder that, in the opinion of the investigators, precluded informed consent or ability to adhere to the study protocol. Written informed consent was obtained before enrolment in the local language (Filipino, Cebuano, Hilagaynon) or English.

### Comorbidity outcome [19]

Comorbidities assessed included: acute undernutrition defined as moderate or severe (body mass index, BMI <17.0 kg/m$^2$) [20]; diabetes, defined as HbA1c >6.5% [21] or currently on a recognised treatment for diabetes; moderate or severe anaemia (haemoglobin <11 g/dL) [22]; hypertension defined as systolic blood pressure (SBP) ≥140 mmHg or a diastolic blood pressure (DBP) of ≥90 mmHg, equivalent to stage 2 hypertension by the 2017 American College of Cardiology and American Heart Association guidelines [23], or currently taking anti-hypertensive medications hypertension; and, HIV (positive HIV screening test or reported existing diagnosis). Severity of raised BP and hypertension was further categorized into elevated BP (systolic BP 120–129 mmHg & diastolic BP <80), stage 1 (systolic BP 130–139 mmHg, or diastolic BP 80–89 mmHg) and stage 2 (as above).

### Data collection

Enrolment took place between 1$^{st}$ August 2018 and 21$^{st}$ February 2020. Trained research nurses interviewed participants, completed all study assessments using structured questionnaires and extracted information recorded on individuals' National TB Program treatment cards, using direct electronic data capture with tablets using Open Data Kit software [24]. Data

were uploaded to a secure server daily. Household food security was assessed and categorized as food secure or mild food insecurity (raw score 0–2), moderate food insecurity (raw score 3–5) and severe food insecurity (raw score 6–9) using the Adapted U.S. Adult Household Food Security Survey Module (US HFSSM) [19, 25].

Research nurses conducted anthropometry, including weight (to the nearest 0.1 kg; Seca 803 Clara Digital Personal Non-Medical Scale) on a flat surface with the patient standing upright and unassisted without shoes. Heights were taken (to nearest 0.5 cm; Seca 216 Mechanical Stadiometer) without shoes or socks, with the patient standing unsupported and positioned fully upright with the lower border of the left orbit and the upper margin of the external auditory meatus horizontal. BMI was categorised for analysis according to WHO criteria; overweight/obese = BMI>25.0 kg/m$^2$; normal = BMI 18.5–25.0, BMI<18.5 = underweight (BMI<18.5 - $\geq$17.0 = mild underweight, BMI <17- $\geq$16.0 = moderate underweight, BMI<16.0 kg/m2 = severe underweight.

Waist and hip circumferences were measured (to the nearest 0.5 cm; Seca 201 measuring tape), midway between the uppermost border of the iliac crest and lower border of costal margin with tape parallel to the floor. Hip circumference was measured at the widest portion of the buttocks with the tape parallel to the floor [26]. Waist-to-hip ratio was calculated by dividing waist circumference by hip circumference. A high waist-to-hip ratio of $\geq$0.85 for females and 0.9 for males was used an indicator of central obesity [26].

Blood pressure was measured twice 5 minutes apart with the participant seated and at rest using an automated blood pressure monitor (Omron HEM-907, Kyoto, Japan). If measurements were $\geq$ 5mmHg apart, a third measure was taken and the average of the 2 closest values used. Fingerprick blood samples were used to obtain haemoglobin (HemoCue 301, Ängelholm, Sweden), HbA1C, (Trinity Biotech) and conduct HIV screening (Standard Diagnostics Bioline HIV-1/2 Ag/Ab Combo Rapid Test kits) for those with unknown status and who provided additional consent.

### NCD health risk behaviour definitions

Participants were asked if they were current smokers or had ever smoked, their average number of cigarettes per day and the number of years smoked; used to calculate the total number of pack years. Participants were asked if they regularly drank alcohol before their current TB diagnosis and if yes, how frequently on average either, daily, weekly or monthly and this was used to number of drinks/yr. For assessing smoking and alcohol use on risk of diabetes or hypertension, these were categorised as 4-level variables; none/never or irregular then total pack-years or total drinks/year divided into tertiles. Participants were also asked if they had changed their smoking or alcohol use since the current diagnosis. Questions related to drug abuse were not asked for political and legal reasons.

### Sample size

The sample size was determined by the primary objective of the cohort, which was to investigate associations between each of diabetes and undernutrition and treatment outcome. A minimum sample size of 800 was estimated to provide 90% power (1–ß) and 5% ($\alpha$ = 0.05) significance to detect associations between moderate or severe undernutrition (BMI<17 kg/m$^2$, odds ratio $\geq$2.1) or diabetes (odds ratio $\geq$2.8) at start of treatment with adverse TB treatment outcomes vs treatment completion or cure, as defined programmatically [27], assuming 10–20% experience an adverse treatment outcome, 25–30% are undernourished and 10–12% are diabetic. Post-hoc sample size calculations were not performed for the analyses reported here.

### Analysis

Characteristics of participants were tabulated overall and by TB treatment regimen (drug sensitive or multi-drug resistant) and by area (Negros Ocidental, Cebu and Manila). Differences in categorical characteristic variables between areas were tested using chi-squared/test. Logistic regression was used for univariable and multivariable analysis of factors associated with each co-morbidity. Multivariable models for each outcome comorbidity were developed using forward step-wise selection of variables in three blocks, starting with socio-demographic characteristics, then nutrition-related variables and finally TB-related variables. Variables in each block were tested for inclusion in order of the most significant in univariable analysis. Selection of a final model for each outcome was based on inclusion of factors associated with an outcome based on a likelihood ratio test (LRT) p-value of ≤0.1 and retained if meeting this criterion after adjustment for other covariates.

### Ethical approval

The study was approved by the Nationally accredited Ethical Review Committee of the Asian Eye Institute, Manila, Philippines (ref 2018–008) and the Ethical Review Committee of Nagasaki University School of Tropical Medicine and Global Health, Japan and the London School of Hygiene & Tropical Medicine, UK (ref 14894).

## Results

Of 6,981 potentially eligible individuals started on either the standard DS-ATT or the WHO shorter MDR regimen at the 13 participating clinics during the enrolment period, 903 persons (12.9%) were enrolled (**S1 Fig & S1 Table in S1 File**). Fewer potentially eligible persons were approached for enrolment in Negros Occidental (7.1%) compared to Cebu (35.6%) and Metro Manila (41.9%). This was due to one very large clinic site in the provincial capital of Negros Occidental, with a large catchment area in this mostly rural province, some of which was outside of our recruitment area. However, there was a similar age and sex distribution between those enrolled and registered and between the sites. The analysis population included 900 participants, (**Fig 1**) two-thirds male with a mean age of 44.7 years (standard deviation, sd 16.4), ranging from 18–87 years. Fifteen percent were initiated on the WHO shorter regimen for drug-resistant TB, with a significantly lower proportion in Metro Manila compared to Cebu and Negros Occidental sites (p<0.001).

Participant characteristics vary across regions (**S2 Table in S1 File**). Participants in Negros Occidental were generally older, less educated, less likely to be employed, have lower incomes but more likely to have health insurance. Food insecurity was highest in Metro Manila (36%), followed by Negros Occidental (25%) and Cebu (19%). Households were larger in Negros Occidental, followed by Cebu, but household density (persons/number of rooms) was higher in Metro Manila. TB-related characteristics also differed significantly by region; bacteriologically confirmed TB was 51% in Cebu and Negros Occidental compared to 41% in Metro Manila. A history of previous TB treatment was more common in Negros Occidental (38%) compared to Cebu (34%) and Metro Manila (28%), with 20% of participants in Negros Occidental reporting one or more household contacts, ever having been previously diagnosed with TB, compared to 8% in Cebu. Participants in Negros Occidental also had a longer duration and number of symptoms at start of treatment, with much higher prevalence of participants reporting current appetite loss, weight loss, night sweats and haemoptysis, possibly indicating more severe disease in these participants.

Socio-demographic and TB-related characteristics did not appear to differ between participants initiated on drug-sensitive (DS-TB) and MDR-TB regimens, apart from a longer duration and number of TB-related symptoms and a higher proportion of participants with a

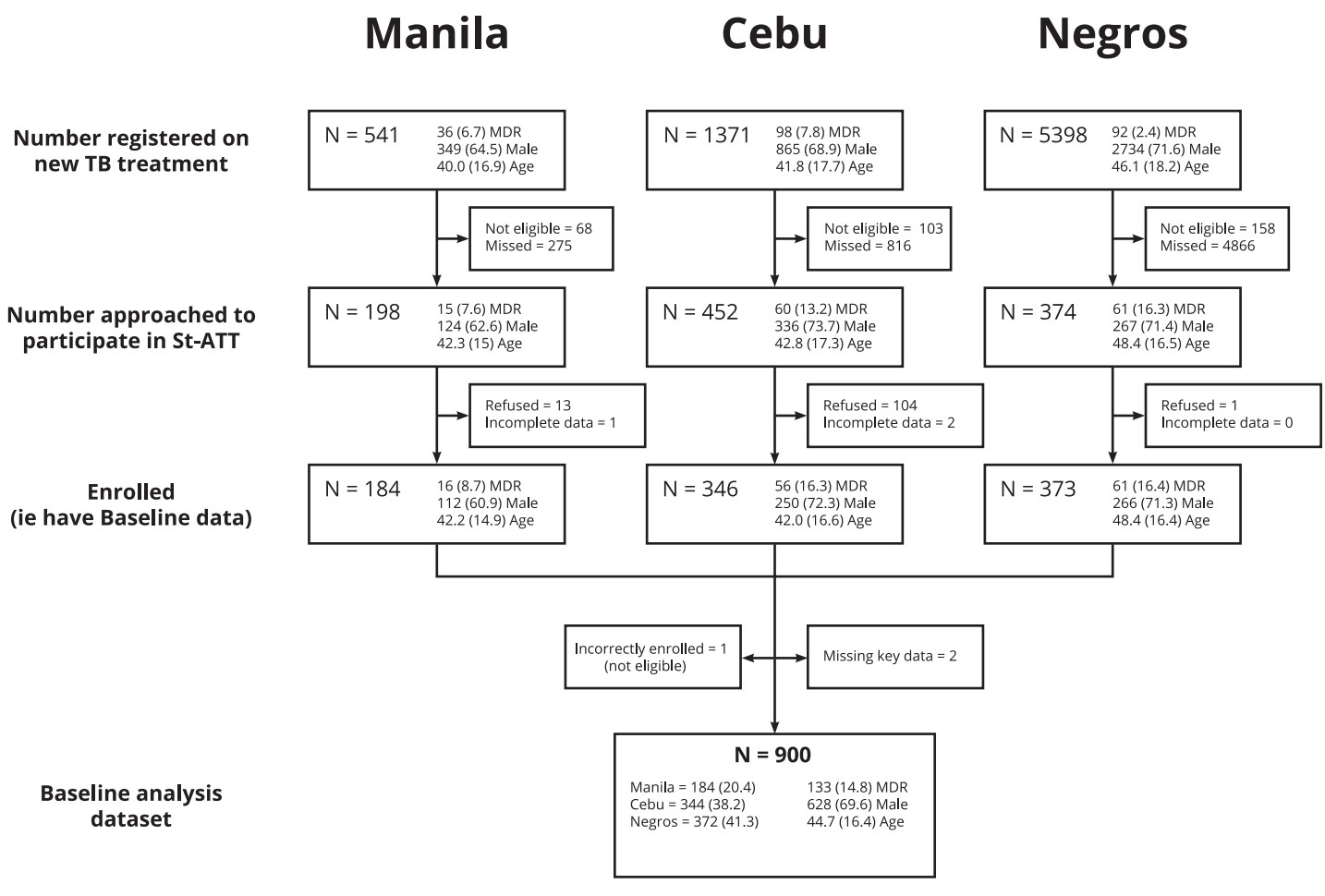

**Fig 1. Flow chart of enrolment.**

history of previous TB treatment (26% vs 75%, p<0.001) in those initiating MDR TB treatment (**Table 1**). All but 1 of the 133 participants on MDR-TB treatment were bacteriologically confirmed and 5.5% of New TB cases were initiated on MDR treatment, ranging from 6.9% in Cebu to 2.5% in Metro Manila (**S3 Table in S1 File**).

## Prevalence of NCD risk factors: Smoking and alcohol behaviours

By self-report, 22% were current smokers and 36% past smokers, with median total pack-years of 10 [interquartile range [IQR] 2.5–22]. More individuals in Negros Occidental were current or ex-smokers (64%, p<0.004 by region) and there were more current smokers in Cebu (28% p<0.001 by region). There was no evidence of different smoking behavior between MDR and DS-TB treatment regimens (**Fig 2**). Amongst current smokers, 90% reported decreasing their usual smoking since their TB diagnosis (183/202) and 83% of ex-smokers may have quit as a result of their TB symptoms or diagnosis (267/303). Overall, one-third of participants reported usually drinking alcohol at least once per month, with more in Cebu and Negros compared to Metro Manila (chi-squared test p = 0.002) but did not differ by MDR vs DS TB treatment regimen (p = 0.555). Frequency of alcohol intake, varied by region and MDR vs DS TB treatment (p<0.001, p = 0.036) (**Fig 2**). Of those that reported regular alcohol consumption, 94% reported decreasing their alcohol intake since diagnosis (293/311).

**Table 1. Socio-demographic and TB-related characteristics of study participants at enrollment.**

| Characteristic | | All (N = 900) | DS-TB (N = 767) | MDR-TB (N = 133) | p-value |
|---|---|---|---|---|---|
| **Socio-demographic characteristics** | | | | | |
| Age group | 18–40 years old | 380 (42.2) | 325 (42.4) | 55 (41.4) | 0.081 |
| | 41–65 years old | 419 (46.6) | 349 (45.5) | 70 (52.6) | |
| | ≥ 65 years old | 101 (11.2) | 93 (12.1) | 8 (6.0) | |
| Marital status | Single | 374 (41.6) | 320 (41.7) | 54 (40.6) | 0.501 |
| | Married | 428 (47.6) | 359 (46.8) | 69 (51.9) | |
| | Divorced/separated/widowed | 98 (10.9) | 88 (11.5) | 10 (7.5) | |
| Education | Primary | 258 (28.7) | 229 (29.9) | 29 (21.8) | 0.333 |
| | Secondary | 420 (46.7) | 350 (45.6) | 70 (52.6) | |
| | Tertiary or vocational | 214 (23.8) | 181 (23.6) | 32 (24.1) | |
| Employed | No | 560 (62.2) | 461 (60.1) | 99 (74.4) | 0.002 |
| | Yes | 339 (37.7) | 305 (39.8) | 34 (25.6) | |
| Family income | Less than 5,000 PHP | 372 (41.3) | 312 (40.7) | 60 (45.1) | 0.934 |
| | 5000–9999 PHP | 243 (27.0) | 211 (27.5) | 32 (24.1) | |
| | 10,000–14,999 PHP | 171 (19.0) | 146 (19.0) | 25 (18.8) | |
| | 15,000–19,999 PHP | 41 (4.6) | 36 (4.7) | 5 (3.8) | |
| | 20,000 PHP or more | 44 (4.9) | 37 (4.8) | 7 (5.3) | |
| | Don't know | 28 (3.1) | 24 (3.1) | 4 (3.0) | |
| Food Insecurity | Food Secure | 675 (75.0) | 581 (75.7) | 94 (70.7) | 0.392 |
| | Moderate Food Insecurity | 169 (18.8) | 141 (18.4) | 28 (21.1) | |
| | Severe Food Insecurity | 56 (6.2) | 45 (5.9) | 11 (8.3) | |
| Have Health Insurance | | 557 (61.9) | 484 (63.1) | 73 (54.9) | 0.072 |
| Median household size (IQR) | Adult (18+) | 2 (1–3) | 2 (1–3) | 2 (1–3) | 0.362 |
| | Young (5–18) | 1 (0–2) | 1 (0–2) | 1 (0–2) | 0.066 |
| | Children (0–5) | 0 (0–1) | 0 (0–1) | 0 (0–1) | 0.863 |
| | Whole household | 3 (2–5) | 3 (2–5) | 4 (2–5) | 0.113 |
| Median Household density (IQR) | | 2.0 (1.4) | 2.0 (1.4) | 2.3 (1.8) | 0.866 |
| Region | Metropolitan Manila | 372 (41.3) | 311 (40.5) | 61 (45.9) | 0.033 |
| | Negros | 344 (38.2) | 288 (37.5) | 56 (42.1) | |
| | Cebu | 184 (20.4) | 168 (21.9) | 16 (12.0) | |
| **TB-related characteristics** | | | | | |
| New Tx or Relapse | New | 598 (66.4) | 565 (73.7) | 33 (24.8) | <0.001 |
| | Relapse/Failure/TALF/PTOU | 302 (33.6) | 202 (26.3) | 100 (75.2) | |
| Basis of diagnosis | Clinical diagnosis | 457 (50.8) | 456 (59.5) | 1 (0.8) | <0.001 |
| | Bacteriologically confirmed | 443 (49.2) | 311 (40.5) | 132 (99.2) | |
| DSSM grade | Negative | 258 (28.7) | 253 (33.0) | 5 (3.8) | <0.001 |
| | 1+ | 65 (7.2) | 64 (8.3) | 1 (0.8) | |
| | 2+ | 30 (3.3) | 25 (3.3) | 5 (3.8) | |
| | ≥3+ | 60 (6.7) | 43 (5.6) | 3 (2.3) | |
| Household TB history | ≥1 HHC ever diagnosed TB | 131 (14.6) | 110 (14.3) | 21 (15.8) | 0.002 |
| Median duration (days) symptoms before start of Tx (IQR) | | 48.0 (30.0–77.0) | 47.0 (30.0–75.0) | 53.0 (28.0–89.0) | 0.094 |
| Current TB symptoms | Cough | 795 (88.3) | 667 (87.0) | 128 (96.2) | <0.001 |
| | Fatigue | 572 (63.6) | 487 (63.5) | 85 (63.9) | <0.001 |
| | Fever | 350 (38.9) | 292 (38.1) | 58 (43.6) | <0.001 |
| | Night sweats | 292 (32.4) | 246 (32.1) | 46 (34.6) | <0.001 |
| | Reduced appetite | 350 (38.9) | 300 (39.1) | 50 (37.6) | <0.001 |
| | Chills | 186 (20.7) | 157 (20.5) | 29 (21.8) | <0.001 |

*(Continued)*

**Table 1.** (Continued)

| Characteristic | | All (N = 900) | DS-TB (N = 767) | MDR-TB (N = 133) | p-value |
|---|---|---|---|---|---|
| | Chest pain | 317 (35.2) | 265 (34.6) | 52 (39.1) | <0.001 |
| | Weight loss | 522 (58.0) | 434 (56.6) | 88 (66.2) | <0.001 |
| | Haemoptysis | 275 (30.6) | 218 (28.4) | 57 (42.9) | <0.001 |
| | Other | 95 (10.6) | 80 (10.4) | 15 (11.3) | <0.001 |

TALF = treatment after loss-to-follow-up; PTOU = previous treatment outcome unknown; DSSM = direct sputum smear microscopy; HHC = household contact;

Tx = treatment; PHP = Philippines Peso ($1 ~ 50 PHP)

Mean Age and % male/female and Multi-drug resistant (MDR-TB)/Drug sensitive (DS-TB) are shown in Fig 1

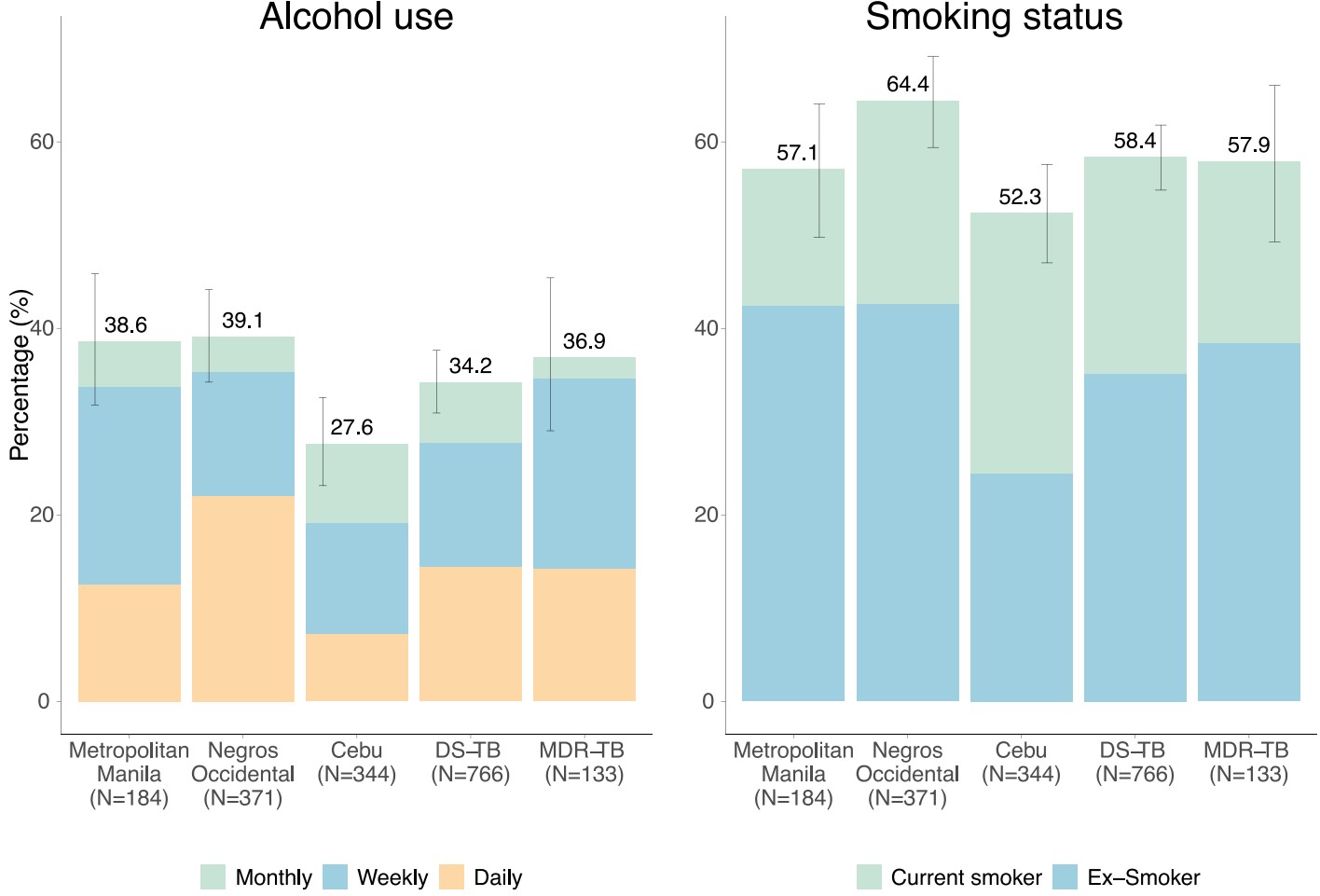

**Fig 2. Prevalence of risk behaviours by region and by drug sensitive or multi-drug resistant TB treatment regimen. Footnotes: Smoking status:** Prevalence of smoking differed significantly by region (chi2, p = 0.004) but not by MDR vs DS TB treatment regimen (chi2, p = 0.92). The relative proportion of current compared to ex-smoking behaviour differed significantly by region (chi2, p<0.001) but not by MDR vs DS TB treatment regimen (chi2, p = 0.31). **Alcohol intake:** Prevalence of reported regular intake of alcohol differed significantly by region (chi2, p = 0.002) but not by MDR vs DS TB treatment regimen (chi2, p = 0.55). The frequency of alcohol intake in those that reported regular drinking differed significantly by region (chi2, p<0.001) and by MDR vs DS TB treatment regimen (chi2, p = 0.036).

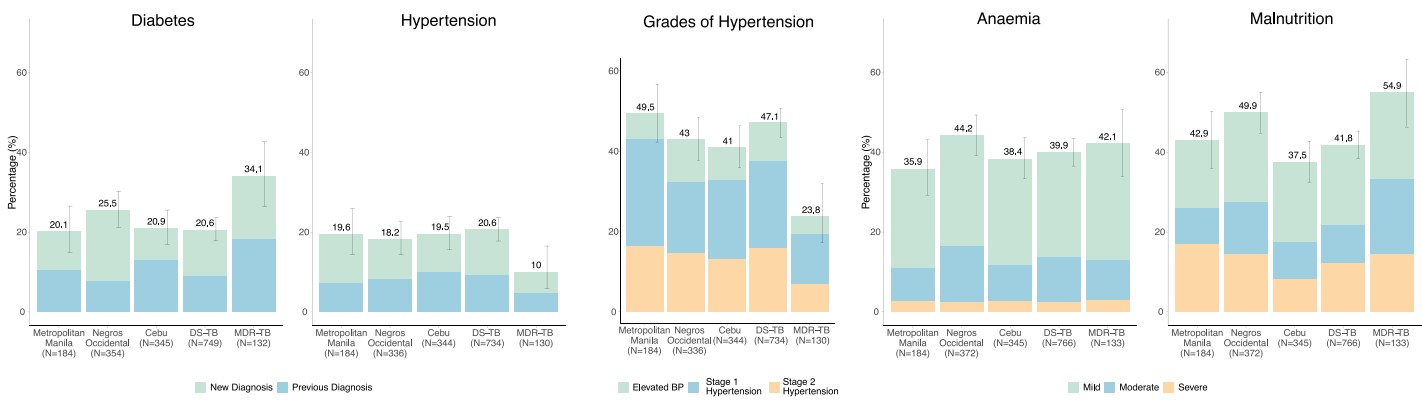

**Fig 3. A**. Prevalence, severity and existing and new diagnoses of diabetes and hypertension comorbidities by region and by drug sensitive or multi-drug resistant TB treatment regimen. **Diabetes:** HbA1c 6.5% or currently taking recognised diabetes medication. Previous diagnosis = reported diagnosis prior to TB diagnosis. Prevalence of diabetes did not differ by region (chi2 p = 0.24) but was higher in MDR vs DS TB treatment (chi2, p = 0.001). The proportion of new *vs.* previous diabetes diagnoses differed significantly by region (chi2, p<0.001) but not by MDR vs DS TB treatment regimen (chi2, p-0.21). **Hypertension:** defined as stage 2 hypertension: Systolic BP>140 mmHg OR diastolic BP>90 mmHg or currently taking recognised anti-hypertensive medication. Previous diagnosis = reported diagnosis prior to TB diagnosis. The prevalence of hypertension did not differ significantly by region (chi2, p = 0.91) but was lower in MDR vs DS TB treatment (chi2, p = 0.005). The proportion of new vs previous hypertension diagnoses did not differ by region or MDR vs DS TB treatment regimen (chi2, p = 0.741; p = 0.280). **Grades of Hypertension:** Elevated BP = elevated systolic BP $\geq$ 120 mmHg < 130 mmHg & normal diastolic BP <80 mmHg; Stage 1 Hypertension = systolic BP $\geq$ 130 mmHg < 140 mmHg OR diastolic BP $\geq$ 80 mmHg < 90 mmHg; Stage 2 hypertension: Systolic BP>140 mmHg or diastolic BP>90 mmHg. The prevalence of elevated BP/hypertension (all grades combined) did not differ significantly by region (chi2, p = 0.170) but did by MDR vs DS TB treatment regimen (chi2, p<0.001). The relative proportion of grades of hypertension (elevated, stage 1, stage 2) did not differ by region or MDR vs DS TB treatment regimen (chi2, p = 0.19, p = 0.84). **B.** Prevalence and severity of malnutrition and anaemia comorbidities by region and by drug sensitive or multi-drug resistant TB treatment regimen. **Footnotes: Undernutrition:** Mild = Body Mass Index (BMI) <18.5–17 kg/m2; Moderate = BMI <17–16 kg/m2; Severe = BMI<16 kg/m2. The prevalence of all grades of undernutrition differed significantly by region (chi2, p = 0.004) and by MDR vs DS TB treatment regimen (chi2, p = 0.005). The relative proportions of grade of undernutrition did not differ by region (chi2, p = 0.085) or by MDR vs DS TB treatment regimen (chi2, p = 0.12). **Anaemia:** Mild = Haemoglobin (Hb) <13.0 [male] <12.0 [female] g/dl—>11.0 g/dl; Moderate Hb $\leq$11.0–8.0 g/dl; Severe Hb <8.0 g/dl. The prevalence of all grades of anaemia did not vary significantly by region or by MDR vs DS TB treatment regimen (chi2, p = 0.49 & p = 0.77).

## Prevalence and characteristics of comorbidities

**Diabetes.** At the start of TB treatment 22.6% (199/881; 95% CI: 19.8–25.4%) had study-defined diabetes (**Fig 3A**). There was no evidence of a difference in prevalence by region, but there was between those on DS-TB vs MDR-TB treatment (**Fig 3A**; 20.6% vs 34.1%; p = 0.001). Of those with diabetes, 45% (90/199) reported a previous clinical diagnosis which differed by region, being highest in Cebu and lowest in Negros Occidental. Of 90 participants with a previous diabetes diagnosis, 49 (54%) reported regular diabetes follow-up visits at a health center or with a doctor, whilst 77 (86%) reported currently taking diabetes medication of whom 68, were taking metformin, glycazide, or in combination with insulin. However, amongst those reporting current diabetic medication, only 23/73 (31.5%) were controlled (HbA1c $\leq$8.0%; [28]) with HbA1c ranging to over 14% in the remaining (**Fig 4**). HbA1c values were higher in previously diagnosed versus newly diagnosed diabetes (median HbA1c% = 9.6% [IQR: 7.5% - 11.6%] *vs.* 7.5% [IQR: 6.7% - 10.8%]); Wilcoxon rank sum test p = 0.007 (**Fig 4**). Although the majority of diabetes cases had normal BMI (117/199, 59%), or were overweight/obese (28/199, 14%), there were 54 (27%) who were malnourished (14 with moderate (BMI 16 - <17 kg/m$^2$) and 15 with severe undernutrition (BMI<16 kg/m$^2$)) (**Fig 4**. When separated by TB treatment regimen, there were no significant differences in HbA1c levels in those who had diabetes or in the ratio of previous to new diagnoses (**S2 Fig**). Overall, there were small increases in HbA1c with increasing age (ß-coefficient = 0.02, p<0.001), but this association was not consistent when assessed in those with diabetes (ß-coefficient = -0.03, p = 0.026) and cases of diabetes were not limited to those in the older age group with 36/199 (18.1%) occurring in participants aged 40 or less, giving rise to a diabetes prevalence of 9.5% (15/380) in this age group. (**S3 Fig**)

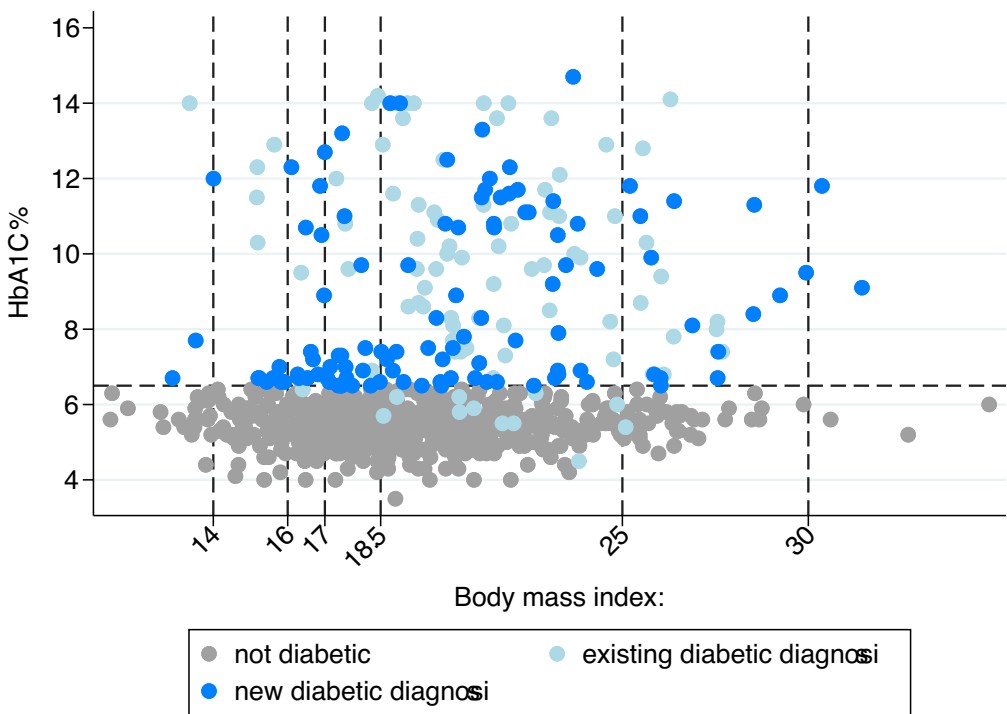

**Fig 4. HbA1c by BMI for new and previous diabetes diagnoses at start of TB treatment. Footnotes:** y-axis: 6.5% cut-off to determine diabetes status. x-axis: BMI cut offs indicate WHO cut-offs of nutritional status; <14 = very severe thinness, <16 severe thinness, <17 = moderate thinness, <18.5 mild thinness, <25 normal, <30 overweight, ≥30 obese.

*Hypertension.* The prevalence of hypertension was 19.0% (164/864; 95% CI: 16.3–21.5%) and did not differ by region (**Fig 3A**) but was lower in those on MDR-TB than DS-TB regimens (**Fig 3A**, p<0.001). The relative proportion of new vs previous hypertension diagnoses did not differ by region or by MDR vs DS TB treatment regimen (**Fig 3A**). When all grades of abnormal BP were considered (elevated BP, stage 1 or stage 2 hypertension by AHA criteria [23]), the overall prevalence increased to 43.6%, (377/864) which did not vary significantly by region but was lower in those enrolled on MDR TB treatment (**Fig 3A**). The relative proportions of grade of abnormal BP did not differ by region or MDR vs DS TB treatment regimen (**Fig 3A**). Forty-six percent (76/164) of those with hypertension reported a previous diagnosis with 70 of these reporting taking a recognised anti-hypertensive medication (losartan and amlodipine being the most common). However, of these only 38/70 (54%) were controlled (BP < 140/90 mmHg) with systolic BP ranging up to 198 mmHg in the remaining (**Fig 5**). There was strong evidence of higher systolic and diastolic BP values in persons with new versus previous hypertension diagnoses (mean SBP = 147.2 mmHg [sd: 15.1] *vs.* 136.9 mmHg [sd: 20.7]; p<0.001; & mean DBP = 89.6 mmHg [sd: 9.7] vs. 81.9 mmHg [sd: 11.9]; p<0.001) (**Fig 5**). Both systolic and diastolic BP were positively associated with age (p<0.001), but cases of study defined hypertension were not limited to older adults, with 17/164 (10.4%) occurring in participants aged 40 years or less (**Fig 5**), giving rise to a hypertension prevalence of 4.5% (17/380) in this age group.

*Undernutrition.* The prevalence of all grades of undernutrition defined as BMI < 18.5 kg/m$^2$ was 43.7% (393/899; 95% CI: 40.-47.0%) and significantly differed by region, being most prevalent in Negros Occidental (49.9%, p = 0.004) and more common in participants enrolled

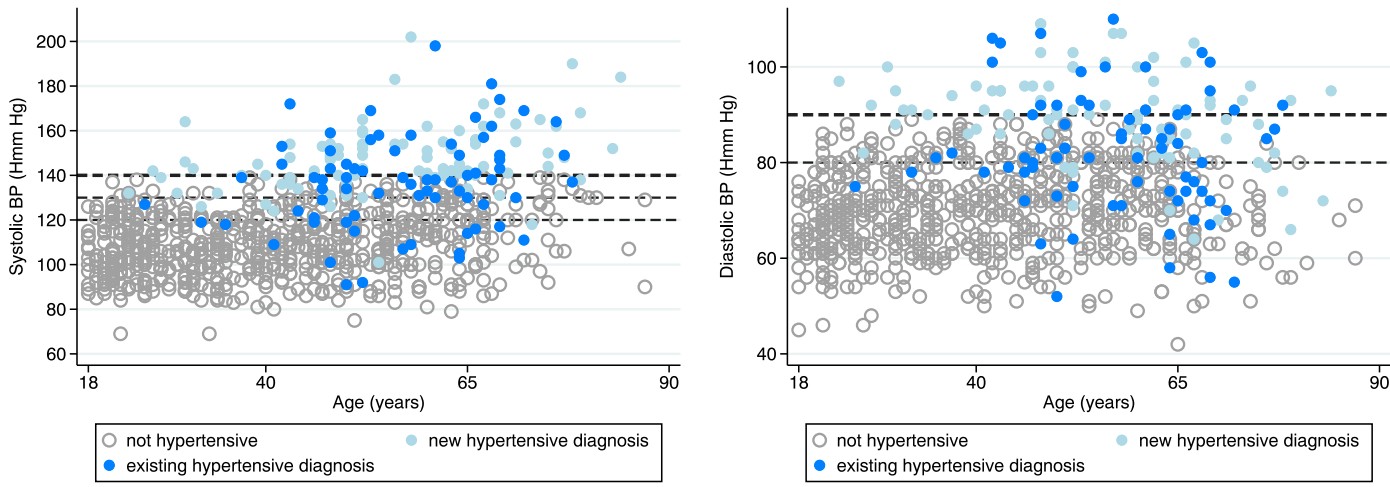

**Fig 5. Systolic and diastolic blood pressure by age in those with and without hypertension and new vs pre-existing diagnoses. Footnotes:**
Hypertension = SPB ≥ 140 mmHg OR DBP ≥ 90 mmHg OR previous diagnosis and currently taking recognised anti-hypertensive medication. Systolic blood pressure (SBP): y-axis: ≥120 mmHg indicates cut-off for elevated SBP; ≥130 mmHg indicates cut off for stage 1 hypertension; ≥140 mmHg indicates cut-off for stage 2 hypertension. Diastolic BP (DBP): y-axis: ≥80 indicates cut-off for elevated DBP; ≥90 indicates cut off for stage 2 hypertension (AHA 2017 criteria [22]).

on MDR compared to DS TB treatment regimens (54.9% vs 41.8%, p = 0.005) (**Fig 3B**). The relative proportion of those who had severe undernutrition (BMI < 16 kg/m2) was possibly higher in Metro Manila (39.2%, p = 0.085) (**Fig 3B**) and there was no evidence of a difference by TB treatment regimen (p = 0.120) (**Fig 3B**). The overall prevalence of moderate or severe undernutrition (BMI < 17.0 kg/m$^2$), selected to define clinically significant co-morbidity was 23.4% (210/899; 95% CI: 20.6–26.1%), which also significantly differed by region (p = 0.004) and by MDR vs DS TB treatment regimen (p = 0.004).

The prevalence of all grades of anaemia (Mild = haemoglobin (Hb) <13.0 [male] <12.0 [female] g/dl—>11.0 g/dl; Moderate Hb ≤11.0–8.0 g/dl; Severe Hb <8.0 g/dl.) was 40.3% (362/899; 95% CI: 37.1–43.5%) and did not differ significantly by region or by MDR vs DS TB treatment regimens, nor did the relative proportion of grades of anaemia (**Fig 3B**). The overall prevalence of moderate or severe anaemia (haemoglobin <11 g/dl), selected to define clinically significant co-morbidity, was 13.5% (121/899; 95% CI: 11-2-15.7%), possibly higher in Negros Occidental (16.5%, p = 0.085) compared to the other regions, but not different by treatment regimen (**Fig 3B**).

### Risk factors for co-morbid conditions

Detailed univariable analysis results are presented for each outcome of diabetes, hypertension, anaemia and undernutrition in **S4-S7 Tables in S1 File**.

**Diabetes.** After adjustment, older age, being married versus single, weight increase or loss in the past 3–6 months, MDR-TB, high waist-to-hip ratio and MDR versus DS treatment regimen were associated with increased odds of diabetes (**Table 2**). Moderate and severe food insecurity compared to none or mild and underweight BMI classification compared to normal were associated with lower odds of diabetes. In the final multivariable model high waist-to-hip ratio, as an indicator of central obesity, accounted for the univariable association observed for overweight or obese by BMI.

**Hypertension.** After adjustment, those aged 65 years and above compared to those aged 41–64 years and those who were overweight or obese compared to normal had increased odds

**Table 2. Multivariable odds ratios (95% confidence intervals) for factors associated with each comorbidity.**

| | | Diabetes[1] (N = 881) | Hypertension[2] (N = 864) | Anaemia (Mod/Severe)[3] (N = 899) | Undernutrition (Mod/Severe)[4] (N = 899) |
|---|---|---|---|---|---|
| **Socio-demographic characteristics** | | | | | |
| Age[3] | 18–40 years old | 1 | **0.13 (0.08–0.23)** | 1 | **2.19 (1.51–3.18)** |
| | 41–64 years old | **2.58 (1.61–4.14)** | 1 | **1.66 (1.06–2.59)** | 1 |
| | 65+ years old | **2.51 (1.31–4.81)** | **2.68 (1.63–4.43)** | **1.91 (1.00–3.54)** | **1.41 (0.82–2.43)** |
| Marital status | Single | 1 | - | - | - |
| | Married | **2.09 (1.31–3.31)** | - | - | - |
| | Divorced/separated | 0.75 (0.23–2.48) | - | - | - |
| | Widowed | 0.85 (0.40–1.84) | - | - | - |
| Education level achieved | Tertiary/vocational | - | - | - | 1 |
| | Secondary | - | - | - | **1.83 (1.14–2.94)** |
| | Primary | - | - | - | **1.85 (1.09–3.13)** |
| Food insecurity level | None/mild | **1** | - | - | 1 |
| | Moderate | **0.49 (0.30–0.82)** | - | - | 1.04 (0.68–1.61) |
| | Severe | **0.83 (0.38–1.81)** | - | - | **2.23 (1.21–4.10)** |
| **Nutrition-related risk factors** | | | | | |
| Weight change in last 3–6 months | No change | 1 | - | - | |
| | Weight increase | 0.92 (0.41–2.18) | - | - | |
| | Weight decrease | **1.97 (1.19–3.27)** | - | - | |
| BMI classification[5] | Normal | 1 | 1 | - | |
| | Underweight | **0.50 (0.33–0.75)** | **0.43 (0.27–0.66)** | - | |
| | Overweight/obese | 1.53 (0.83–2.82) | **3.11 (1.73–5.62)** | - | |
| BMI classification with undernutrition classification[6] | Normal | - | - | 1 | |
| | Mild underweight | - | - | **2.11 (1.26–3.54)** | |
| | Moderate underweight | - | - | **3.34 (1.86–5.99)** | |
| | Severe underweight | - | - | **2.49 (1.39–4.47)** | |
| | Overweight/obese | - | - | 0.64 (0.22–1.88) | |
| Appetite related food intake in past month vs pre-TB symptoms | No change | - | 1 | 1 | 1 |
| | Moderate/severe decrease | - | **0.52 (0.33–0.81)** | **1.65 (1.06–2.57)** | **1.66 (1.14–2.42)** |
| | Increase | - | 1.12 (0.68–1.87) | 1.28 (0.70–2.32) | 1.04 (0.61–1.77) |
| **TB-related risk factors** | | | | | |
| Waist-to-hip ratio[7] | Normal | 1 | - | - | - |
| | High | **2.41 (1.64–3.56)** | - | - | - |
| TB treatment regimen | DS | 1 | 1 | - | - |
| | MDR | **2.48 (1.55–3.95)** | **0.55 (0.39–0.78)** | - | - |
| HIV status | Negative | - | - | 1 | - |
| | Unknown | - | - | 1.48 (0.93–2.35) | - |
| | Positive | - | - | **4.51 (1.01–20.10)** | - |

(*Continued*)

**Table 2.** (Continued)

| | | Diabetes[1] (N = 881) | Hypertension[2] (N = 864) | Anaemia (Mod/Severe)[3] (N = 899) | Undernutrition (Mod/Severe)[4] (N = 899) |
|---|---|---|---|---|---|
| TB type | New diagnosis | - | | - | 1 |
| | Relapse/Failure | - | | - | **1.98 (1.40–2.80)** |
| Duration of TB symptoms | <1 month | - | - | - | 1 |
| | 1–2 months | - | - | - | 1.29 (0.83–2.00) |
| | 2–3 months | - | - | - | **1.88 (1.15–3.06)** |
| | >3 months | - | - | - | **1.92 (1.17–3.13)** |

Retention of covariates in multivariable models based on a likelihood ratio test p-value≤0.1 when comparing adjusted models with and without the covariate. Bold type: effects with a Wald test p-value <0.05. Baseline levels of age category varied in models depending on data scarcity and model fit.

**1:** Diabetes: HbA1c ≥6.5% or on diabetes medication

**2:** Hypertension: Systolic blood pressure ≥140 mmHg or diastolic blood pressure ≥90 mm Hg

**3:** Anaemia: moderate or severe = haemoglobin /<11g/dL

**4:** Undernutrition: moderate or severe BMI<17 kg/m$^2$

**5:** BMI<18.5 = underweight, normal = BMI 18.5–25.0, overweight/obese = BMI>25.0 kg/m2

**6:** BMI 18.5–25.0 kg/m2 = normal; BMI<18.5 - ≥17.0 = mild underweight, BMI <17-≥16.0 = moderate underweight, BMI<16.0 kg/m2 = severe underweight

**7:** High waist-to-hip ratio ≥0.85 for females and 0.9 for males.

**Likelihood ratio test p-values for model covariates by comorbidity** (Bonferroni corrected p-value = 0.05/19 = 0.003)

<u>Diabetes:</u> age group p<0.001; marital status p = 0.001; food insecurity p = 0.018; weight loss p = 0.009; BMI p = 0.001; waist-to-hip ratio p<0.001; TB regimen p<0.001

<u>Hypertension:</u> age group p<0.001; food intake p = 0.004; BMI class p<0.001; regimen p = 0.012

<u>Anaemia:</u> age group p = 0.041; BMI class p<0.001; food intake p = 0.085; HIV p = 0.072

<u>Undernutrition:</u> age group p <0.001; education p = 0.027; food security p = 0.039; food intake p = 0.019; weight change p<0.001; TB type p<0.001; symptom duration p = 0.0.

of hypertension. Persons aged 18–40 *vs.* 41–64 years, who were underweight versus normal, who experienced a moderate or severe decrease in appetite related food intake and on a MDR vs DS treatment regimen had reduced odds of hypertension (**Table 2**).

**Anaemia.** After adjustment, older age groups, those with moderate and severe decreases in recent, appetite-related food intake and mild, moderate or severe undernutrition compared to normal had increased odds of anemia. HIV status was retained in the model with a non-significantly increased odds in those with unknown status and in the few individuals with known positive status (**Table 2**).

**Undernutrition.** After adjustment, those aged 18–40 and more than 65 years compared to 41–64 years, those with lower levels of education and those with a history of previous TB treatment and those with longer duration of TB symptoms before start of current treatment had increased odds of undernutrition. MDR versus DS-TB treatment regimen was no longer associated after adjustment for a history of previous TB treatment. Severe food insecurity, recent weight loss and recent, appetite-related reduced food intake were also associated with increased odds of undernutrition (**Table 2**).

## Multi-morbidity

Out of 847 individuals with complete data for the 4 co-morbidities, 496 (58.6%) had at least one co-morbidity and 127 (15.0%) individuals had 2 and 18 (2.1%) individuals had 3 or more (**Fig 6**). The most common combination was diabetes with hypertension (57, 6.7%), followed by moderate or severe undernutrition with moderate or severe anaemia in 45 individuals (5.3%) (**Fig 6A**). The pattern of co-morbidities appeared to differ in those on MDR-TB

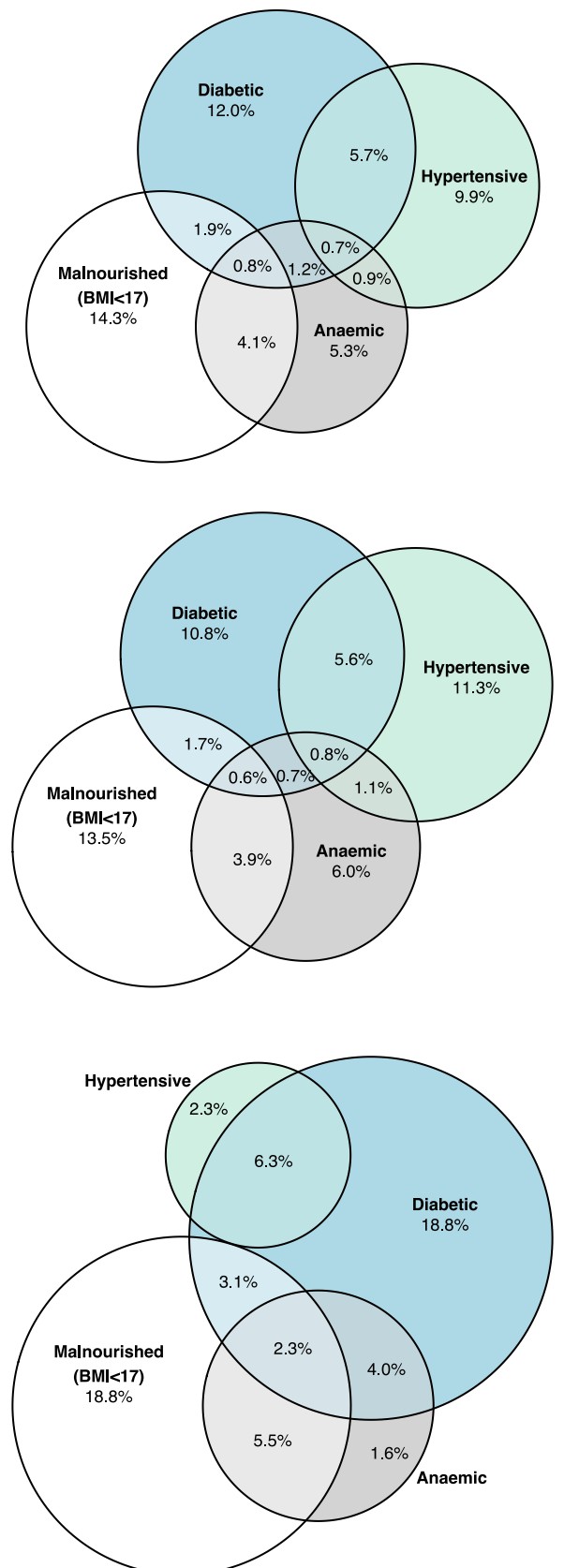

**Fig 6.** Venn diagrams of comorbidities in those with complete data for all 4 comorbidities for [A] All participants, N = 847; [B] participants initiating drug sensitive TB treatment regimens, N = 718; [C] participants initiating multi drug resistant TB treatment regimens, N = 129. **Footnotes: Diabetes:** HbA1c≥6.5% OR on current recognised medication; **Hypertension:** Stage 2, SPB ≥ 140 mmHg OR DBP ≥ 90 mmHg OR previous diagnosis and currently taking recognised anti-hypertensive medication; **Anaemia:** Moderate or severe haemoglobin ≤11.0 g/dL; Malnutrition: moderate or severe, BMI <17 kg/m².

treatment regimens with 22.4% (29/129) of individuals with 2 or more co-morbidities and the most common combination of co-morbidities being undernutrition and anaemia (n = 10; 7.7%) compared to 16.2% (116/718) in those on DS-TB treatment with the most common combination of co-morbidities being diabetes and hypertension in 45 individuals (6.8%) (**Fig 6B & 6C**).

## Discussion

The results of this study summarise the burden and predictors of NCD co-morbidities and their inter-relationships in Filipino persons initiating TB treatment. To our knowledge the current data represents the most comprehensive data available for the Philippines, and more globally for undernutrition and anaemia amongst persons with TB. The WHO End TB strategy includes the need to meet the complete healthcare needs of persons with TB. This is important both for the potential to improve, directly or indirectly, TB treatment outcomes, e.g. from food supplementation for undernutrition or better management of diabetes), but also from a health equity perspective and to protect others in NCD programmes from being exposed to persons with potentially infectious TB.

Overall, nearly 60% of the study population had 1 or more co-morbid conditions, but with relatively little overlap between them. Of the four co-morbidities, moderate or severe undernutrition was the most common, closely followed by diabetes, both affecting just under a quarter of participants, followed by hypertension and lastly anaemia. When mild grades of undernutrition or anaemia were considered the prevalence of these conditions roughly doubled to 44% and 40% of the study population. Just over half of those with diabetes and hypertension were previously undiagnosed, whilst of those that were already on treatment more than half had poor control of hyperglycemia or BP.

The prevalence of diabetes in persons just starting TB treatment was considerably higher than in our previous cross-sectional study in persons at different stages of TB treatment (22.6%, vs. 9.1%) [17]. Some of this difference may be due to TB-induced hyperglycemia causing elevated Hba1c that resolves over the course of TB treatment [28]. However, if TB-induced hyperglycaemia were the explanation we would expect to observe: i) longer duration of TB symptoms before start of treatment associated with higher HbA1c levels; and/or, ii) a higher proportion of newly diagnosed vs pre-existing diabetes to be observed in this study, neither of which we observed [17]. Alternatively, a greater loss to follow-up during treatment of those with comorbid TB-DM might explain the higher proportion of DM in this study. Investigating the proportion of people with elevated HbA1c who self-resolve with TB treatment will be addressed as a secondary objective in the main longitudinal St-ATT cohort, and effects of diabetes and glycemic control on TB treatment outcomes is a primary objective.

Similar to our previous findings, the data suggest that only a minority of those on diabetes medication achieved glycemic control in the period preceding their TB diagnosis, (31.5%) even using the pragmatic <8% HbA1c cut-off as recently recommended for TB-DM [28]. This is not dissimilar to that reported in Filipino persons without TB, receiving standard care at health facilities of a similar level (37% HbA1c < 7.0%; N = 164) [29]. The odds of diabetes were approximately 2.5 times greater in those initiating an MDR-TB treatment regimen, but

this did not appear to be mediated through any association with previous history of TB history, similar to our previous finding but in contrast with the multi-country TANDEM study [30]. Furthermore, there was no indication in an increase in the relative proportion of new vs, pre-existing DM diagnoses between DS- and MDR-TB treatment regimens, which might be expected if MDR-TB was associated with increased TB-induced hyperglycaemia.

Central obesity as indicated by a raised waist-to-hip ratio was strongly associated with diabetes suggesting that this may be a better predictor and potentially useful screening tool for TB-DM in this population [17, 31]. Low BMI was protective, but cannot exclude DM, as just over a quarter of all DM cases were undernourished (BMI<18.5). Importantly, even after adjustment for sample weighting for location, age and sex the prevalence of diabetes in these TB patients is considerably greater at 16.6% (95% CI:13.1–20.1%) than the estimated national adult prevalence of 7.1% [11].

The prevalence of hypertension was lower than that of diabetes, and in contrast to diabetes was significantly lower in those initiating MDR-TB treatment regimens and independent of the large effects of age and under- or overweight status. There is currently no evidence to suggest that hypertension is associated with increased risk of TB [32]. The proportion of participants with both diabetes and hypertension was surprisingly low, although this proportion was higher in those initiating MDR- than DS-TB treatment. It is possible that in this population the relatively younger age of those with diabetes compared to the general population may underlie this observation. Although, the degree of overlap may have been considerably greater if we had used a lower BP cut-off in our hypertension definition. Similar to diabetes, reported use of anti-hypertensive medication did not correlate well with lower BP measurements.

Anaemia was the only condition observed to be associated with HIV status, whilst undernutrition was the only comorbid condition to be independently associated with any TB-related exposures, including increased odds of a previous history of TB and longer duration of TB-related symptoms before starting TB treatment. The prevalence of both these conditions were slightly higher than our previous observations, which is to be expected as TB treatment should contribute to some resolution of these conditions; this is something we will explore within the longitudinal data analysis of this cohort.

Data from the three regions suggest undernutrition, diabetes, hypertension and anemia are important co-morbidities across all regions and in both MDR and DS-TB, but that diabetes and hypertension tend to cluster together in those who are better nourished, whilst undernutrition and anaemia cluster in those who are less educated and poorer–as indicated by household food security. There is some evidence to suggest that diabetes may increase the risk of MDR-TB, rather than more TB-induced hyperglycemia occurring in MDR-TB. Although undernutrition was more common in MDR-TB than DS-TB this was accounted for by the higher proportion of those with a previous history of TB in those on MDR-TB treatment. Unfortunately, without further data it is not possible to determine the relative causal direction of this association. Effective TB treatment alone may not be enough to support nutritional recovery in patient populations at high risk of TB associated catastrophic costs. Failure of nutritional recovery is also known to be an indicator of inadequate treatment, although the mechanisms are still to be elucidated.

Somewhat surprisingly, there was no evidence of association between smoking and alcohol use and any of the comorbidities assessed. However, just under a quarter of participants were current smokers, although most reported decreased smoking as a result of their TB symptoms or diagnosis and many others had recently quit. The proportion of those that remain as non-smokers when TB-symptoms improve is unknown. This demonstrates the need for smoking cessation support to be included within TB programmes in the Philippines.

The data from this study builds on our previously reported cross-sectional study, [17] providing more definitive data on TB-comorbidity at TB treatment initiation, including systematic assessment of hypertension.

### Study strengths, limitations and further research

A strength of this study is the systematic measurement of comorbidities and description of their overlap and management in a large, well-described cohort of persons with TB representative of those initiating new treatment regimens in public facilities across a range of urban, peri-urban and rural settings in a high TB burden country in Asia, in which there is a different pattern of NCDs and HIV compared to most African countries. Study limitations include the high proportion of participants with unknown HIV status due to refusal of testing, although HIV and HIV co-infection in TB in the Philippines is known to be low. Due to logistical considerations, it was not possible to randomly select TB-DOTs clinics or patients from the clinics and this could have led to selection bias. However, despite the smaller proportion of patients enrolled from one of the very large clinics, age and sex distributions of the enrolled vs registered participants were reassuringly similar. Another increasingly recognized comorbidity of depression and anxiety was not included here as this was assessed in a subset and will be reported separately. Finally, causality between risk factors is difficult to infer from the multivariable analyses conducted and requires further research. How these comorbid conditions change over the course of TB treatment and their effect on TB treatment outcomes will be reported within the main findings of the cohort.

### Conclusions

More than half of persons initiating TB treatment regimens have one or more comorbid conditions requiring management as part of patient centered care to improve TB-related treatment outcomes and quality of life. The planning of such services needs better data to design services to improve both the diagnosis and management of these. Nutritional advice and management should be a core component of diabetes and hypertension management [33] but is often missing from programmes due to human resource limitations. Health staff trained in nutritional counselling or management are usually limited to those working in maternal and child health programmes, focusing on undernutrition. We propose that there is a need for an increased cadre of health workers trained in nutrition who could work across integrated programmes of TB and NCDs which would require training in managing nutrition-related chronic diseases and infection-related undernutrition of adults (HIV or TB). Further research is needed on how to optimize management in persons with multimorbidities of which nutrition should be considered a core component.

### Supporting information

**S1 Fig. Map of study sites in the three regions of the Philippines.** Footnotes: Red crosses indicate Tertiary health facilities enrolling persons initiating multi-drug resistant TB treatment regimens. Blue crosses represent primary health centers.
(EPS)

**S2 Fig. HbA1c for new and previous diabetes diagnoses at start of TB treatment by TB treatment regimen.** Footnotes: y-axis: 6.5% cut-off to determine diabetes status. X-axis: BMI cut-offs indicate WHO cut-offs of nutritional status; <14 = very severe thinness, <16 severe thinness, <17 = moderate thinness, <18.5 mild thinness, <25 normal, <30 overweight, ≥30 obese. Median HbA1c in those with study defined diabetes on DS-TB vs MDR TB treatment

regimens = 8.7 [IQR: 6.8–11.1] vs 8.5 [IQR: 6.8–11.6]; p = 0.71.
(EPS)

**S3 Fig. HbA1c by age in new and previous diabetes diagnoses.** Footnotes: y-axis: 6.5% cut-off to determine diabetes status.
(EPS)

**S1 File.**
(DOCX)

## Acknowledgments

The authors would like to thank the participants and their families, staff at the health centers and TB-DOTS clinics and our team of research nurses Cristelyn Alvarez, Clarinda Berido, Michelle Caballero, Bliss Caraig, Paul Ian Flores, Romil Jeffrey Juson, Ann Lustresano, Trivon Opinion, Michelle Saavaedra and Ares Verde for their hard work and dedication to the study and the wellbeing of the participants.

## Author Contributions

**Conceptualization:** Sharon E. Cox, Juan A. Solon.

**Data curation:** Benjamin N. Faguer, Julius P. Ferrer, Shuichi J. Suzuki, Mitsuki Koh, Farzana Ferdous.

**Formal analysis:** Sharon E. Cox, Tansy Edwards, Benjamin N. Faguer.

**Funding acquisition:** Sharon E. Cox.

**Investigation:** Sharon E. Cox, Julius P. Ferrer, Shuichi J. Suzuki, Mitsuki Koh, Naomi R. Saludar, Anna-Marie C. G. Garfin, Juan A. Solon.

**Methodology:** Sharon E. Cox, Tansy Edwards, Benjamin N. Faguer, Juan A. Solon.

**Project administration:** Sharon E. Cox, Juan A. Solon.

**Supervision:** Sharon E. Cox, Tansy Edwards, Shuichi J. Suzuki, Mitsuki Koh, Mary C. Castro.

**Visualization:** Benjamin N. Faguer.

**Writing – original draft:** Sharon E. Cox.

**Writing – review & editing:** Sharon E. Cox, Tansy Edwards, Benjamin N. Faguer, Julius P. Ferrer, Shuichi J. Suzuki, Mitsuki Koh, Farzana Ferdous, Naomi R. Saludar, Anna-Marie C. G. Garfin, Mary C. Castro, Juan A. Solon.

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
