## [Decision Letter · Decision Letter 0]

7 Jul 2021

 PGPH-D-21-00013 Patterns of non-communicable comorbidities at start of tuberculosis treatment in three regions of the Philippines: The St-ATT cohort PLOS Global Public Health

Dear Dr. Cox,

Thank you for submitting your manuscript to PLOS Global Public Health. After careful consideration, we feel that it has merit but does not fully meet PLOS Global Public Health’s publication criteria as it currently stands. Therefore, we invite you to submit a revised version of the manuscript that addresses the points raised during the review process. 

We look forward to receiving your revised manuscript.

Kind regards,

Siyan Yi, MD, MHSc, PhD

Academic Editor

Journal Requirements:

Additional Editor Comments (if provided):

In addition to reviewers’ comments, please follow STROBE checklist for cross-sectional studies to ensure that all minimum required information is included in the manuscript. Please fill in the form and submit it as a supplementary document. Below are some examples for your consideration, which are not exhaustive.

Abstract:

– A structured abstract with background, methods, results, and conclusions would improve its readability.

– Please include a clear study objective at the end of the background.

– In methods, please provide brief info on study design, period, sites, and sampling methods.

– To accommodate other essential information, please reduce descriptive results.

Introduction

– Please break down the first paragraph into 2–3 smaller paragraphs.

Methods

– In my understanding, this analysis used cross-section data, not the prospective cohort study.

– The description of the minimum required sample size estimation is not precise. The section should come earlier, perhaps with ‘Participants.’

– Suggest ‘Ethical approval’ be replaced by ‘Ethical considerations’ that also include other aspects of ethics, not only ethical approvals (privacy and confidentiality protection, consent to participate in the study, compensation, risks and benefits, etc.).

Reviewers' comments:

Reviewer's Responses to Questions

**Comments to the Author**

1. Does this manuscript meet PLOS Climate’s publication criteria? Is the manuscript technically sound, and do the data support the conclusions? The manuscript must describe methodologically and ethically rigorous research with conclusions that are appropriately drawn based on the data presented.

Reviewer #1: Partly

2. Has the statistical analysis been performed appropriately and rigorously?

Reviewer #1: Yes

3. Have the authors made all data underlying the findings in their manuscript fully available (please refer to the Data Availability Statement at the start of the manuscript PDF file)?

Reviewer #1: Yes

4. Is the manuscript presented in an intelligible fashion and written in standard English?

PLOS Climate does not copyedit accepted manuscripts, so the language in submitted articles must be clear, correct, and unambiguous. Any typographical or grammatical errors should be corrected at revision, so please note any specific errors here.

Reviewer #1: Yes

5. Review Comments to the Author

Reviewer #1: Review

The authors present important data on the pattern of multi-morbidity among adult patients with both drug susceptible and multi drug resistant tuberculosis in the Philippines. Whilst the subject is very important and there is a lack of published data, there are several key questions and suggestions for improvement that I believe are required prior to consideration for publication.

1. The aim of the study is not explicitly stated. The description of the pattern of comorbidity is not sufficient.

2. How were patients from treatment centers selected? Whilst the authors state “all” that were enrolled on treatment, there are only 900 from 6,981 (12.9%) total patients. This small percentage opens the study results to selection bias and is a significant limitation. Whether these results are representative of the total patients relies on how included patients may have been different to non-included patients.

3. Why were non pulmonary TB patients excluded from the study?

4. The sample size justification is not clear. The authors state that it was calculated by “the primary objective of the cohort”. However, per my first point, this is not clear.

5. Table 1 (and other results presented) are given by enrollment region. Whilst relevant locally, unless the regions provide the reader with an important message by virtue of their representation and applicability to other settings, I suggest that this is changed. It would be more helpful, for example to present the data as drug susceptible versus multi drug resistant tuberculosis. This would be more applicable to other settings and provide more scope for discussion of the implications of the results.

6. What are the implications of these results? The authors do not sufficiently explain why the diagnosis of multi morbidity at TB diagnosis is important and what implications it has for TB programs.

7. For example, they demonstrate that 68% of patients have poorly controlled diabetes mellitus (HbA1c >8%). It may be possible therefore, that better glycaemic control through x intervention at TB diagnosis results in better treatment outcomes or less post TB sequelae.

8. Are TB treatment outcomes known for this cohort? It would be incredibly important to know if the outcomes were worse among specific co-morbidities.

6. PLOS authors have the option to publish the peer review history of their article (what does this mean?). If published, this will include your full peer review and any attached files.

**Do you want your identity to be public for this peer review?** For information about this choice, including consent withdrawal, please see our Privacy Policy.

Reviewer #1: No

**Comments to the Author**

1. Does this manuscript meet PLOS Global Public Health’s publication criteria? Is the manuscript technically sound, and do the data support the conclusions? The manuscript must describe methodologically and ethically rigorous research with conclusions that are appropriately drawn based on the data presented.

Reviewer #2: Yes

2. Has the statistical analysis been performed appropriately and rigorously?

Reviewer #2: Yes

3. Have the authors made all data underlying the findings in their manuscript fully available (please refer to the Data Availability Statement at the start of the manuscript PDF file)?

Reviewer #2: Yes

4. Is the manuscript presented in an intelligible fashion and written in standard English?

Reviewer #2: Yes

5. Review Comments to the Author

Reviewer #2: Introduction

The paper is generally well-written. However, I suggest that the paragraph on Page 4 be divided into 2 paragraphs, at least. A single paragraph should cover one page.

Please write the acronyms for DS-TB and MDR-TB in full when first used.

Methods

Study design: While the overall study may be prospective, this paper presents cross-sectional data making this a cross-sectional study. Please amend accordingly.

Line 78: Please write the acronym DOT in full.

Setting: What were the criteria for the selection of the TB-DOTS clinics i.e. why were these specific clinics selected for inclusion in this study?

Overweight/obesity, underweight, and raised waist-to-hip ratio are all mentioned in the Results section and need to be defined in the Methods, please.

Please include a definition of food insecurity.

Lines 118-120: Reference is made to hip circumference measurement, but this is not described.

Line 131” ‘calculate the’ should come before ‘number’

Data collection: How did fieldworkers enroll participants i.e. select from the potentially eligible participants? For example, using random tables, etc.

Results

The study dates (1st August 2018 and 21st February 2020) belong in the Methods, please.

Table 1:

For your international reader, please write PHP in full in the footnote and the equivalent of 1 PHP to the USD.

Please write the following acronyms in full: ALF/PTOU

Line 236: Why was a cut-point of 8% and not 7% used for diabetes control? Please include the appropriate international/standard/WHO reference.

Lines 279-280: ‘were considered’ is repeated.

Please be consistent with the use of either blood pressure or BP in the text – do not use both.

Line 313: ‘all grades of anaemia’ – please provide the haemoglobin cut-point and the relevant reference

Discussion

Lines 411-414 need rephrasing for better clarity, please. I suggest splitting into 2 sentences for easier understanding.

Lines 407-432: This paragraph is difficult to follow. I suggest dividing into 3-4 paragraphs with shorter sentences for ease of reading.

Lines 440-441: Please amend and remove the emphasis of including systolic blood pressure of 120-129 mmHg as being potentially classified as hypertension.

Lines 446-450; 458-462, 493-497: Please divide into at least 2 sentences each.

Line 469: Please include the reference.

Line 473: Please remove ‘prospective’ – this data is from baseline analyses of a prospective study i.e. cross-sectional assessments were done in this study.

6. PLOS authors have the option to publish the peer review history of their article (what does this mean?). If published, this will include your full peer review and any attached files.

**Do you want your identity to be public for this peer review?** For information about this choice, including consent withdrawal, please see our Privacy Policy.

Reviewer #2: No

---

## [Decision Letter · Decision Letter 1]

20 Oct 2021

Patterns of non-communicable comorbidities at start of tuberculosis treatment in three regions of the Philippines: The St-ATT cohort

PGPH-D-21-00013R1

Dear Dr. Cox,

We're pleased to inform you that your manuscript has been judged scientifically suitable for publication and will be formally accepted for publication once it meets all outstanding technical requirements.

Within one week, you'll receive an e-mail detailing the required amendments. When these have been addressed, you'll receive a formal acceptance letter and your manuscript will be scheduled for publication.

An invoice for payment will follow shortly after the formal acceptance. To ensure an efficient process, please log into Editorial Manager at https://www.editorialmanager.com/pgph/ click the 'Update My Information' link at the top of the page, and double check that your user information is up-to-date. If you have any billing related questions, please contact our Author Billing department directly at authorbilling@plos.org.

Kind regards,

Siyan Yi, MD, MHSc, PhD

Academic Editor

Additional Editor Comments (optional):

Reviewers' comments:

Reviewer's Responses to Questions

**Comments to the Author**

1. If the authors have adequately addressed your comments raised in a previous round of review and you feel that this manuscript is now acceptable for publication, you may indicate that here to bypass the “Comments to the Author” section, enter your conflict of interest statement in the “Confidential to Editor” section, and submit your "Accept" recommendation.

Reviewer #1: All comments have been addressed

Reviewer #2: All comments have been addressed

2. Does this manuscript meet PLOS Global Public Health’s publication criteria? Is the manuscript technically sound, and do the data support the conclusions? The manuscript must describe methodologically and ethically rigorous research with conclusions that are appropriately drawn based on the data presented.

Reviewer #1: Yes

Reviewer #2: (No Response)

3. Has the statistical analysis been performed appropriately and rigorously?

Reviewer #1: Yes

Reviewer #2: (No Response)

4. Have the authors made all data underlying the findings in their manuscript fully available (please refer to the Data Availability Statement at the start of the manuscript PDF file)?

Reviewer #1: Yes

Reviewer #2: (No Response)

5. Is the manuscript presented in an intelligible fashion and written in standard English?

Reviewer #1: Yes

Reviewer #2: (No Response)

6. Review Comments to the Author

Reviewer #1: The authors have addressed my questions and made appropriate changes to improve the manuscript.

I have nothing further to add.

Reviewer #2: Lines 42 and 44: Please remove ‘disease’ after TB

Lines 366-367, 377-379: Please improve syntax; there is unnecessary use of the words ‘those’/’those with’

7. PLOS authors have the option to publish the peer review history of their article (what does this mean?). If published, this will include your full peer review and any attached files.

**Do you want your identity to be public for this peer review?** For information about this choice, including consent withdrawal, please see our Privacy Policy.

Reviewer #1: No

Reviewer #2: No
